# Sleep quality of college students in Fujian and its influencing factors: A cross-sectional study

**Fuzhi Liu**[1], **Dandan Zhu**[1], **Lifeng Deng**[1], **Yanyan Zhao**[2], **Yangjia Chen**[1], **Zhuote Tu**[1]*

**1** Department of Preventive Medicine, School of Health, Quanzhou Medical College, Quanzhou, China,
**2** Department of Nursing, Quanzhou First Hospital, Quanzhou, China

* 75582394@qq.com

## Abstract

### Aim

This study aims to investigate college students' sleep quality, explore the factors influencing it, and provide data support for further studies.

### Methods

College students in Fujian Province were chosen as the study sample using snowball sampling. Data was gathered from the participants through the utilization of a self-designed personal questionnaire, the Pittsburgh Sleep Quality Index(PSQI) scale, and the Mobile Phone Addiction Index (MPAI) scale. Binary logistic regression is utilized to assess the sleep quality of college students and identify risk factors.

### Results

A total of 971 participants were included in this study. The mean total PSQI score was $4.52\pm3.17$. There were 310 students with poor sleep quality and the detection rate was 32.0%. The multivariate logistic regression analysis showed that smoking (OR(Odds ratio):1.832(1.076,3.118)), electronic product addiction(OR:2.861(1.928,4.246)), personal history of acute illness(OR:2.328(1.671,3.244)) were identified as independent risk factors.In turn, education (OR:0.594(0.456,0,772)) and parents without sleep problems (OR:0.533(0.361,0.787)) were protective factors.

### Conclusion

College students in Fujian have some sleep problems. We should pay attention to the relationship between smoking, electronic product addiction, personal history of acute illness and sleep quality. Health policymakers should consider these factors to improve college students' sleep quality.

**Data availability statement:** All relevant data are within the manuscript and its Supporting Information files.

**Funding:** This study was granted by the Natural Science Foundation of Quanzhou (Grant No.2021N124S PI: Fuzhi Liu) The funders had no role in study design, data collection and analysis, decision to publish, or preparation of the manuscript.

**Competing interests:** The authors have declared that no competing interests exist.

**Abbreviation:** PSQI, Pittsburgh Sleep Quality Index; MPAI, Mobile Phone Addiction Index; OR, Odds Ratio; SD, Standard Deviation; KMO, Kaiser-Meyer-Olkin; VIF, Variance Inflation Factor; AOR, Adjust Odds Ratio.

# Introduction

Sleep is a necessary process of body activities. Sleep causes physiological changes in all body systems and organs due to alterations in the autonomic and somatic nervous systems [1]. Sleep is crucial for good physical and mental health. Research shows poor sleep quality affects many adults, including university students, with estimates of prevalence ranging from 6% to 94% [2]. As we all know, College students face various pressures, including academics, socializing, admission to higher education, and employment. Sleep problems among college students are on the rise and vary in nature, as shown by numerous studies [3–5].

Sleep problems will be used to refer generally to any combination of acute or chronic problems with prolonged sleep onset latency (SOL), excessive wake after sleep onset (WASO), short total sleep time (TST), low sleep efficiency (SE), or poor sleep quality based on subjective and/or objective assessments [6]. Sleep problems does not refer to symptoms of specific clinical sleep disorders [7]. Poor sleep quality can harm college students' health, leading to physiological and psychological changes and increasing the risk of related diseases. The literature reported that there were close associations between poor sleep status and illness, including hypertension [8], diabetes [9], and alzheimer's disease dementia [10]. Accumulating evidence shows that poor sleep status is also related to mental health, especially anxiety [11] and depression [12]. The profound influence of sleep problems on college students has attracted wide attention from students and society.

The Pittsburgh Sleep Quality Index (PSQI) is a widely used self-report questionnaire for clinicians and researchers to assess different aspects of sleep problems [13]. The questionnaire has 19 items that generate 7 component scores, including subjective sleep quality, sleep latency, sleep duration, habitual sleep efficiency, sleep disturbances, use of sleep medications, and daytime dysfunction.The PSQI has been validated in diverse populations, such as insomnia, psychiatric patients, and health workers [14–16].

A systematic review found that 18.5% of adolescents worldwide have sleep quality problems, significantly higher than the general population's reported rate of 7.4% [17]. Existing studies have reported that various factors can affect the sleep quality of college students, including individual characteristics, behavioral and lifestyle factors, as well as physical and mental health factors. A cross-sectional study revealed that poor sleep quality was prevalent among women, medical students, and sophomore students [18]. A systematic review showed that dietary habits, including dietary patterns or individual healthy/unhealthy foods, are associated with sleep quality [19]. The PSQI evaluation in U.S. college students revealed a strong link with depression, anxiety, and sleep quality [20]. Furthermore, research conducted in various regions has revealed that distinct student demographics exhibit differing prevalence rates of sleep disorders, which are influenced by a range of factors. A cross-sectional survey conducted in Afghanistan indicated that the prevalence of sleep disorders among participants was 46.4%. The identified predictors included being aged 13-16 years, having lower class grade, having an uneducated fathers, being of low income, and parental unemployment [21]. Additionally, a study focusing on Muslim students reported a prevalence rate of 29.1% for sleep disorders, with findings suggesting that age, economic status, and parental employment were significantly associated with the occurrence of sleep disturbances among the respondents [22].

In summary, these studies provide evidence supporting the prevalence of poor sleep quality among college students, which is influenced by various factors. It is important to acknowledge that the quality of sleep may differ across various geographical regions, and there is a scarcity of research on sleep quality specifically in Fujian Province. Over the past few years, college students have exhibited the highest usage rates of electronic devices, potentially resulting in electronic device addiction and subsequent sleep quality issues. In particular, the survey

was conducted in December 2022, a period characterized by a heightened incidence of the COVID. During this time, the predominant mode of instruction for students was online teaching, which resulted in an increase in their Internet usage duration. Consequently, we employed the PSQI questionnaire to evaluate the sleep quality and associated risk factors among individuals residing in Fujian Province, China.

## Materials and methods

### Study population and sampling

This cross-sectional survey was conducted among college students, undergraduate students and postgraduate students in Fujian province from December 1 to 31, 2022. Inclusion criteria: All participants provided their informed consent to take part in this study. Exclusion criteria: 1) Participants were received treatment for sleep disorders (e. g., insomnia) in the past year. 2) Participants had a family history of sleep disorders.

This study utilized a snowball sampling approach to collect data from participants. The survey was conducted using the Wenjuanxing, which is the most commonly used online questionnaire platform. The structured questionnaire has been developed to generate links on the platform in an automated manner. These links can be efficiently disseminated to selected participants via prominent communication channels like QQ (a widely utilized instant messaging service in China) and WeChat. By leveraging these channels, additional respondents can be reached, thereby facilitating the gradual expansion of the sample pool. Before completing the questionnaire, participants were asked to read and confirm their understanding of an informed consent form provided at the beginning of the survey. All questionnaires were reviewed and checked daily by automatic proofreading of Wenjuanxing and supervisors to ensure the quality of data and its completeness. There are 26 questionnaires with logic errors that cannot be revisited. Finally, we were able to obtain a satisfactory number of completed questionnaires, amounting to 971, which adequately fulfills the requirements of our research.

### Questionnaires

The questionnaire consisted of two parts, including a general survey questionnaire and the Pittsburgh Sleep Quality Index (PSQI) questionnaire.

**General Survey Questionnaire.** This part included the primary demographic data of age, sex, education, major and the influence factors such as tobacco, alcohol, physical activity, lifestyle, personal history of illness, and others.

**Mobile Phone Addiction Index.** Electronic product addiction was measured to Mobile Phone Addiction Index (MPAI) [23] which included 17 self-reported items. Each item is scored on a five-point Likert-type scale(not at all = 1, rarely = 2 occasionally = 3, often = 4, always = 5). The assessment tool comprises a maximum score of 85, where elevated scores indicate a greater severity of mobile phone misuse. A score of less than 35 signifies the absence of mobile phone addiction, while a score of 35 or higher indicates the presence of mobile phone addiction. This scale is widely used to investigate the addiction to mobile phones among teenagers and college students. It was developed by Leung and has good reliability and validity. In this study, the Cronbach alpha value was 0.757.

**Pittsburgh Sleep Quality Index questionnaire.** This standard measurement scale was used to assess sleep quality over the previous month. The scores range from 0 to 3 in each component, and when summed, produce the total PSQI scores ranging from 0 to 21. Scores > 5 indicate poor sleep quality, and ≤ 5 indicate good sleep quality [13,24]. The Cronbach α for the PSQI was previously reported as 0.83 in the original validation study. Furthermore, this questionnaire has been extensively utilized in various regions, including China [25,26]. In this

research, the Cronbach α value was 0.754, the Kaiser-Meyer-Olkin (KMO) value was 0.816, and the P value of the Bartlett sphericity test less than 0.05.

### Ethical considerations

This study was approved by the Ethics Committee of Quanzhou First Hospital (Quan Yi Lun 2022248). Have an informed consent form at the beginning of the questionnaire, and participants need to read this section and click "Yes, I participate in." We received 997 questionnaires, and all agreed to participate in this study. The responses and submissions of the questionnaire do not necessitate the inclusion of personally identifiable confidential information; furthermore, during the data analysis phase, personal data, including IP addresses, will remain undisclosed. Throughout the snowball sampling process, the confidentiality of information pertaining to both the initial sample and potential participants will be strictly maintained.All procedures performed in studies involving human participants have been performed following the Declaration of Helsinki.

### Statistical analysis

All data were pre-processed with Microsoft Excel and processed with SPSS 24.0. Measurement data were expressed as means and standard deviation(SD). Count data were described by rate. Student's t-test, Analysis of Variance, and Chi-square test were used for the data. Logistic regression analysis was conducted to determine the impact of variables on sleep quality. We utilize the R software and apply Firth's Logistic regression method to mitigate the bias associated with maximum likelihood estimation when dealing with sparse data [27]. For all tests, $p \leq 0.05$ was taken as the significance level.

## Results

### Demographic statistics

The study surveyed 971 participants. The mean age of the participants was 20.65 years old. 315 were male and 656 were female. 412 were medical students and 559 were non-medical students. The college, undergraduate and postgraduate numbers were 515, 416, and 40, respectively. In monthly living expenses, there are 464 students mainly concentrated in 1000-1500 yuan (Table 1).

### The PSQI general scores and PSQI component scores of the studied population

The investigation showed that the average total score of PSQI was 4.52 ± 3.17. The parameters evaluated in the study included subjective sleep quality, sleep latency, sleep duration, habitual sleep efficiency, sleep disturbances, use of sleep medications, and scores indicating daytime dysfunction (with mean values of 0.99 ± 0.75, 1.10 ± 0.98, 0.58 ± 0.80, 0.46 ± 0.82, 0.86 ± 0.63, 0.10 ± 0.43, and 0.07 ± 0.29, respectively), as shown in Table 2.

### Detection and distribution of sleep quality among college students

In this study, students were divided into two groups based on their PSQI score: good sleep quality (PSQI score ≤ 5) and poor sleep quality (PSQI score > 5). Out of 310 students, 32.0% had sleep disorders. The researchers used the Chi-square test to analyze risk factors at a 5% significance level.

**Table 1. Demographic characteristics of college students.**

| Variables | Group | Number(n) | Rate(%) |
|---|---|---|---|
| Age (years old) | <20 | 541 | 55.7 |
| | ≥20 | 430 | 44.3 |
| Sex | male | 315 | 32.4 |
| | female | 656 | 67.6 |
| Major | medical | 412 | 42.4 |
| | non-medical | 559 | 57.6 |
| Education | college | 515 | 53.1 |
| | undergraduate | 416 | 42.8 |
| | postgraduate | 40 | 4.1 |
| Monthly living expenses(yuan) | less than1000 | 92 | 9.5 |
| | from 1000 to 1500 | 464 | 47.8 |
| | from 1500 to 2000 | 291 | 30.0 |
| | from 2000 to 2500 | 57 | 5.9 |
| | more than 2500 | 67 | 6.8 |

**Table 2. The PSQI general scores and PSQI component scores of the studied population.**

| Indices | Mean | SD |
|---|---|---|
| PSQI general scores | 4.52 | 3.17 |
| Subjective sleep quality | 0.99 | 0.75 |
| Sleep latency | 1.10 | 0.98 |
| Sleep duration | 0.58 | 0.80 |
| Habitual sleep efficiency | 0.46 | 0.82 |
| Sleep disturbances | 0.86 | 0.63 |
| Use of sleep medications | 0.10 | 0.43 |
| Daytime dysfunction | 0.07 | 0.29 |

## Sleep quality of college students with different demographics

The Chi-square test analyzed the general characteristics of the research object, including age, sex, education, major, among others. The result display that education (P = 0.005) is statistically significant using Chi-square test of linear trend (Table 3).

## Sleep quality of college students with different lifestyles and behaviors

The study showed that tobacco, midnight snack, electronic product addiction and eating habits are statistically significant (Table 4).

## Sleep quality of college students with other influencing factors

The study found significant statistical correlations between personal history of acute illness, chronic disease, and parental sleep problems (Table 5).

Binary logistic regression analyzed factors affecting college students' sleep quality.

According to the above data analysis, the influencing factors of sleep quality were screened and assigned. Sleep quality is the dependent variable, and the independent variables selected are the influencing factors with alpha less than 0.20 discovered in this study. In addition, the selection of independent variables also comes from the influencing factors of college students'

**Table 3. Comparison of sleep quality among college students with different demographic characteristics.**

| Variable | Group | Good sleep quality (n,%) | Poor sleep quality (n,%) | ² | P |
|---|---|---|---|---|---|
| Age(year) | <20 | 381(57.6) | 160(51.6) | 3.107 | 0.078 |
| | ≥20 | 280(42.4) | 150(48.4) | | |
| Sex | male | 210(31.8) | 105(33.9) | 0.425 | 0.514 |
| | female | 451(68.2) | 205(66.1) | | |
| Major | medical | 271(41.0) | 141(45.5) | 1.738 | 0.187 |
| | non-medical | 390(59.0) | 169(54.5) | | |
| Education | college | 330(49.9) | 185(59.7) | 7.769 | 0.005* |
| | undergraduate | 301(45.5) | 115(37.1) | | |
| | postgraduate | 30(4.6) | 10(3.2) | | |
| Monthly living expenses(yuan) | less than1000 | 59(8.9) | 33(10.6) | 1.174 | 0.882 |
| | from 1000 to 1500 | 321(48.6) | 143(46.1) | | |
| | from 1500 to 2000 | 196(29.7) | 95(30.7) | | |
| | from 2000 to 2500 | 38(5.7) | 19(6.1) | | |
| | more than 2500 | 47(7.1) | 20(6.5) | | |

* indicate p < 0.05..

**Table 4. Comparison of sleep quality among college students with different lifestyles and behavior.**

| Variables | Group | Good sleep quality(n,%) | Poor sleep quality(n,%) | ² | P |
|---|---|---|---|---|---|
| Tobacco | yes | 41(6.2) | 34(11.0) | 6.723 | 0.010* |
| | no | 620(93.8) | 276(89.0) | | |
| Alcohol | yes | 70(10.6) | 42(13.5) | 1.810 | 0.179 |
| | no | 591(89.4) | 268(86.5) | | |
| Beverage | yes | 475(71.9) | 237(76.5) | 2.274 | 0.132 |
| | no | 186(28.1) | 73(23.5) | | |
| Midnight snack | yes | 295(44.6) | 161(51.9) | 4.523 | 0.033* |
| | no | 366(55.4) | 149(48.1) | | |
| Electronic product addiction | yes | 88(13.3) | 83(26.8) | 26.354 | <0.001* |
| | no | 573(86.7) | 227(73.2) | | |
| Eating habits | sweet | 157(23.8) | 63(20.3) | 12.986 | 0.024* |
| | sour | 38(5.7) | 11(3.5) | | |
| | piquancy | 208(31.5) | 92(29.7) | | |
| | saline | 52(7.9) | 39(12.6) | | |
| | light taste | 194(29.3) | 92(29.7) | | |
| | oil | 12(1.8) | 13(4.2) | | |
| Physical activity | never | 108(16.3) | 61(19.7) | 4.367 | 0.225 |
| | sometimes | 380(57.5) | 169(54.5) | | |
| | 1-2 times a week | 76(11.5) | 44(14.2) | | |
| | at least 3 times a week | 97(14.7) | 36(11.6) | | |

* indicate p < 0.05.

sleep quality reported in previous literature but with no statistical differences in this study. Multivariate analysis was conducted using Enter logistic regression analysis to screen variables. Collinearity diagnosis showed no collinearity among covariates, with tolerance above 0.7 and variance inflation factor (VIF) below 3. Variable details can be seen in Table 6.

**Table 5. Comparison of sleep quality among college students with other influencing factors.**

| Variables | Group | Good sleep quality(n,%) | Poor sleep quality(n,%) | 2 | P |
|---|---|---|---|---|---|
| Personal history of acute illness | yes | 113(17.1) | 103(33.2) | 31.745 | <0.001* |
| | no | 548(82.9) | 207(66.8) | | |
| Personal history of chronic disease | yes | 25(3.8) | 27(8.7) | 10.109 | 0.001* |
| | no | 636(96.2) | 283(91.3) | | |
| Parental sleep problems | only one | 122(18.5) | 77(24.8) | 24.400 | <0.001* |
| | none | 290(43.9) | 86(27.7) | | |
| | both | 24(3.6) | 19(6.2) | | |
| | unclear | 225(34.0) | 128(41.3) | | |

* indicate p < 0.05..

**Table 6. Variables assignment of the questionnaire.**

| Variable | Definition |
|---|---|
| Age (year) | 0=<20, 1=≥20 |
| Sex | 0 = male, 1 = female |
| Education | 1 = college, 2 = undergraduate, 3 = postgraduate |
| Major | 0 = non-medical, 1 = medical |
| Monthly living expenses (yuan) | 1 = less than1000, 2 = from 1000 to 1500, 3 = from 1500 to 2000, 4 = from 2000 to 2500, 5 = more than 2500 |
| Tobacco | 0 = no, 1 = yes |
| Alcohol | 0 = no, 1 = yes |
| Beverage | 0 = no, 1 = yes |
| Eating habits | 1 = sweet, 2 = sour, 3 = piquancy, 4 = saline, 5 = light taste, 6 = oil. This variable sets the dummy value. |
| Electronic product addiction | 0 = no, 1 = yes |
| Personal history of acute illness | 0 = no, 1 = yes |
| Personal history of chronic disease | 0 = no, 1 = yes |
| Parental sleep problems | 1 = none, 2 = only one, 3 = both, 4 = unclear. This variable sets the dummy value. |

In the logistic analysis, the variables of age and sex were included as covariates to control for their potential confounding effects. Through the utilization of binary logistic regression analysis, it has been confirmed that there exists an association between sleep quality and various factors. These factors include education (OR = 0.594, P < 0.001), tobacco (OR = 1.832, p = 0.026), electronic product addiction (OR = 2.861, p < 0.001), personal history of acute illness (OR = 2.328, p < 0.001), and parents without sleep problems (OR = 0.533, p = 0.002) (as presented in Table 7).

## Discussion

Based on the established cut-off point (PSQI > 5), the average global PSQI score of university students in this study was found to be $4.52 \pm 3.17$. Furthermore, it was observed that 32% of the participants exhibited poor sleep quality. This article was similar to findings from a study involving Chinese college students, which reported a global Pittsburgh Sleep Quality Index (PSQI) score of $4.51 \pm 2.52$ and a prevalence rate of poor sleep quality at 31.0% [28]. This

**Table 7. Binary logistic regression analysis of the influencing factors of sleep quality in college students.**

| Variables | B | Std. | Wald | P | OR | 95% CI | |
|---|---|---|---|---|---|---|---|
| | | | | | | lower | upper |
| Education | -0.521 | 0.134 | 15.087 | <0.001 | 0.594 | 0.456 | 0.772 |
| Tobacco | 0.605 | 0.271 | 4.976 | 0.026 | 1.832 | 1.076 | 3.118 |
| Electronic product addiction | 1.051 | 0.201 | 27.236 | <0.001 | 2.861 | 1.928 | 4.246 |
| Personal history of acute illness | 0.845 | 0.169 | 24.920 | <0.001 | 2.328 | 1.671 | 3.244 |
| Parents without sleep problems | -0.629 | 0.199 | 9.982 | 0.002 | 0.533 | 0.361 | 0.787 |
| Sex | -0.100 | 0.164 | 0.373 | 0.542 | 0.905 | 0.656 | 1.248 |
| Age | 0.243 | 0.149 | 2.675 | 0.102 | 1.276 | 0.953 | 1.707 |

Note: Age and sex were adjustment factors.

incidence is comparable to those observed in Brazil(35.3%) [29] and New Zealand(37.2) [30]. and, a comprehensive meta-analysis conducted in more than 76 studies involving 112 939 university students in China found that the overall pooled prevalence of sleep disturbances was 25.7% [31]. In addition, our finding is lower than the prevalence of poor sleep quality in a university sample in Saudi Arabia (83%)and the United Kingdom(65%), with the mean global PSQI score being 8.0 ± 3.0 and 6.89 ± 3.03, respectively [32,33]. However, The result of this paper is higher than another study of university students in Iran (14.5%) [34] China (14.8%) [35] and USA(17.1%) [36]. Consequently, the prevalence of sleep quality problems among college students is a matter of concern. The PQSI scores and the prevalence of poor sleep suggest that these phenomena exhibit regional variability, which may be attributable to factors such as the study design employed (cross-sectional, cohort, or case-control), the characteristics of participants across different studies, or divergent sleep and wakefulness patterns. This observation underscores the necessity for further research to investigate potential underlying differences.

It is well known that sleep time becomes shorter, more fragmented, and of poorer quality with advancing age [37]. A recent study has reported that normative aging is linked to a decline in the capacity to initiate and sustain sleep [38]. The research conducted revealed that college students who were 20 years old and above exhibited a notably greater sleep duration (P = 0.031) and experienced more sleep disturbances (P < 0.001) in comparison to their counterparts who were below 20 years of age. Nevertheless, the remaining scores did not exhibit any statistical significance. The results of this study indicate that the sleep quality of older college students is comparatively lower than that of their younger counterparts. The age of our study subjects was 20.65 ± 1.40. This implies that the occurrence of this phenomenon is not attributed to the natural process of aging, but rather to external environmental factors, such as the increased stressors associated with pursuing higher education or engaging in employment. Hence, it might be imperative to implement interventions aimed at enhancing the sleep quality of elderly college students.

Numerous studies have demonstrated that women experience lower sleep quality compared to men [39,40]. Specifically, the present study found that subjective sleep quality (P = 0.012), sleep latency (P = 0.007), and sleep disturbances (P < 0.001) were more prevalent among female participants compared to their male counterparts. Overall, females have poorer sleep quality than males. This may be attributed to women's emotional sensitivity, which renders them more susceptible to external factors, as well as hormonal fluctuations related to menstrual cycles [41].

The findings of this study suggest a negative correlation between educational attainment and the prevalence of sleep disorders, indicating that individuals with higher levels of

education are more inclined to assimilate health-related information, diminish their tobacco use, and limit their reliance on electronic devices, ultimately leading to improved sleep quality. Prior research has demonstrated that peer education can significantly enhance the health literacy of individuals within a peer group, enabling them to discern between favorable and unfavorable mental health conditions, identify potential risks, and implement effective protective strategies [42]

Tobacco use behavior was associated with poor sleep.The present study utilized a multiple logistic regression model to examine the relationship between smoking and sleep quality. The results indicated that smoking (OR = 1.832) was identified as an independent risk factor for poor sleep quality. This finding is consistent with a survey conducted among Chinese individuals, which also reported higher odds (OR = 1.468) of poor sleep quality among cigarette smokers [43]. Another survey suggested that smoking by itself is not associated with sleep quality [44]. The impact of smoking on sleep is predominantly ascribed to the stimulatory properties of nicotine [45]. We can see that smoking is particularly deleterious to sleep quality. Encouraging smoking cessation as a means to improve sleep quality could serve as a positive motivator for student smokers to quit.

Electronic product addiction has an impact on the sleep quality of college students [46–48]. The prevalence of electronic product addiction was 17.6% in this study, which was lower than 34.8% reported by other surveys [49]. This article showed that electronic product addiction (OR = 2.861) is an independent factor of sleep quality. The findings have been established elsewhere in the literature. There is a study on Chinese college students indicated that sleep quality was significantly associated with> four years of smartphone use (OR = 3.38),> five hours of daily smartphone use (OR = 2.19) [50]. College students use their mobile phones mainly for social interaction, playing games and watching videos. It has been reported that utilizing mobile phones for calling and sending text messages after bedtime is linked to sleep disturbances [51]. The overuse of mobile phones prior to bedtime has been found to have adverse effects on students' sleep quality. Specifically, it can delay the onset of deep sleep and reduce the overall duration of this stage of sleep.

Diseases have been found to have a direct impact on sleep patterns. Furthermore, certain chronic and severe illnesses can lead to psychological distress, which in turn can indirectly affect an individual's sleep quality. There was reported in the literature that the number of diseases they had been diagnosed with and atopic dermatitis were positively associated with the sleep quality of students [52]. In another study of college students,overweight participants had a higher AOR of short sleep than normal weight participants (AOR = 1.52); and obese participants had a higher AORs of both short and long sleep (> 9 h/night) (AOR = 1.67; AOR = 1.79, respectively) [53]. The present study utilized multiple logistic regression analysis to investigate the relationship between personal history of acute illness and sleep quality. The results indicated that personal history of acute illness (OR = 2.328) was an independent risk factor for poor sleep quality. These findings suggest that disease among college students may significantly impact their sleep quality.

In addition to the results of the current study, various factors have been identified as influencing sleep quality. Existing literature indicates that additional types of factors may also exert an impact. A cross-sectional survey employing multivariate analysis conducted in Afghanistan revealed that high income was negatively associated with sleep quality (AOR = 4.132, p = 0.002), while working night shifts (AOR = 0.288, p < 0.001) and experiencing a traumatic event within the past month (AOR = 0.504, p = 0.007) were positively correlated with sleep quality [54].

## Conclusion

College students in Fujian have poor sleep quality (32%) and a mean global PSQI score was $4.52 \pm 3.17$. Smoking, electronic product addiction, personal history of acute illness are linked to sleep quality. Health policymakers should address these factors to improve college students' sleep quality.

## Limitations and recommendations

Our research has several limitations:1) The cross-sectional survey conducted in this study does not establish a definitive causal relationship between sleep quality and its influencing factors. Therefore, additional research is required to further investigate this matter. It is important to note that the data for this study was collected using online questionnaires, which may introduce sampling errors. Furthermore, the reliance on self-reported data in the questionnaire may lead to reporting bias, particularly concerning subjective variables such as smoking and alcohol consumption. 2) It is important to acknowledge that various factors, such as final exam pressure, the emotional state and personality of the participants, may have influenced the current results. It is possible that these factors were not taken into consideration, which could potentially lead to different outcomes. 3) The research did not employ a diagnostic instrument to identify sleep disorders, but rather focused on identifying symptoms. 4) Owing to the presence of sampling error, the sample composition of this study differs from previous studies, notably with a high proportion of medical students, thereby influencing the generalizability of the study findings.

Based on this study, the following suggestions are proposed: 1. Enhance health education initiatives, promote regular physical activity, and implement effective exercise programs to decrease the incidence of acute illnesses, while also establishing feasible smoking cessation initiatives. 2. Address the issue of electronic dependency by emphasizing the importance of students' engagement with information resources and fostering a conducive learning environment for the use of electronic devices.

## Supporting information

**S1 Table. The differences between PSQI general and component scores among ages.**
(DOCX)

**S2 Table. The differences between PSQI general and component scores among sex.**
(DOCX)

**S3 Table. The differences between PSQI general and component scores among education.**
(DOCX)

**S4 Table. The differences between PSQI general and component scores among major.**
(DOCX)

**S5 Table. Tolerance and variance inflation factors for different variables.**
(DOCX)

**S1 Data. Source data.**
(XLSX)

## Acknowledgments

We thank the participated students in the study.We also acknowledge the reviewers and editors for viewing our work.

## Author contributions

**Conceptualization:** Yangjia Chen.

**Data curation:** Fuzhi Liu, Dandan Zhu, Yanyan Zhao.

**Formal analysis:** Fuzhi Liu, Dandan Zhu, Lifeng Deng, Yanyan Zhao.

**Funding acquisition:** Fuzhi Liu.

**Investigation:** Fuzhi Liu, Dandan Zhu, Lifeng Deng.

**Methodology:** Dandan Zhu.

**Project administration:** Fuzhi Liu, Dandan Zhu.

**Resources:** Yangjia Chen, Zhuote Tu.

**Software:** Lifeng Deng.

**Supervision:** Zhuote Tu.

**Writing – original draft:** Fuzhi Liu, Dandan Zhu, Lifeng Deng, Yanyan Zhao, Yangjia Chen, Zhuote Tu.

**Writing – review & editing:** Fuzhi Liu, Dandan Zhu, Lifeng Deng, Yanyan Zhao, Yangjia Chen, Zhuote Tu.

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
