## [Decision Letter · Decision Letter 0]

19 Dec 2024

PONE-D-24-42285Sleep quality of college students in Fujian and its influencing factors: a cross-sectional studyPLOS ONE

Dear Dr. Liu,

Thank you for submitting your manuscript to PLOS ONE. After careful consideration, we feel that it has merit but does not fully meet PLOS ONE’s publication criteria as it currently stands. Therefore, we invite you to submit a revised version of the manuscript that addresses the points raised during the review process.

We look forward to receiving your revised manuscript.

Kind regards,

Ahmad Neyazi

Academic Editor

PLOS ONE

3. Thank you for stating the following financial disclosure:  [This study was granted by the Natural Science Foundation of Quanzhou (Grant No.2021N124S PI: Fuzhi Liu)].  Please state what role the funders took in the study.  If the funders had no role, please state: "The funders had no role in study design, data collection and analysis, decision to publish, or preparation of the manuscript." If this statement is not correct you must amend it as needed. Please include this amended Role of Funder statement in your cover letter; we will change the online submission form on your behalf.

4. We note that your Data Availability Statement is currently as follows: [All relevant data are within the manuscript and its Supporting Information files.] Please confirm at this time whether or not your submission contains all raw data required to replicate the results of your study. Authors must share the “minimal data set” for their submission. PLOS defines the minimal data set to consist of the data required to replicate all study findings reported in the article, as well as related metadata and methods (https://journals.plos.org/plosone/s/data-availability#loc-minimal-data-set-definition).

Additional Editor Comments (if provided):

Reviewers' comments:

Reviewer's Responses to Questions

**Comments to the Author**

1. Is the manuscript technically sound, and do the data support the conclusions?

Reviewer #1: Yes

Reviewer #2: Yes

2. Has the statistical analysis been performed appropriately and rigorously? 

Reviewer #1: Yes

Reviewer #2: Yes

3. Have the authors made all data underlying the findings in their manuscript fully available?

Reviewer #1: Yes

Reviewer #2: Yes

4. Is the manuscript presented in an intelligible fashion and written in standard English?

Reviewer #1: Yes

Reviewer #2: Yes

5. Review Comments to the Author

Reviewer #1: The manuscript is well-written and addresses a critical topic. However, there are areas where improvements can enhance its clarity, depth, and overall impact. Below are specific comments categorized by section.

Introduction

1. Broaden the discussion on the importance of sleep quality by incorporating a global perspective and referencing recent studies.

2. Strengthen the rationale for focusing on Fujian Province, particularly the relevance of electronic product addiction in this region.

3. Include the following studies on adolescent sleep quality to enrich the introduction:

DOI: 10.1016/j.sleepe.2024.100102

DOI: 10.1016/j.sleepe.2024.100075

Methodology

1. Provide a more detailed justification for using snowball sampling, considering its limitations in terms of representativeness and potential biases.

2. Explain why the Mobile Phone Addiction Index (MPAI) was selected and how it is particularly suited for this study population.

3. Include examples of psychometric validation of the Pittsburgh Sleep Quality Index (PSQI) from other countries (e.g., DOI: 10.1007/s44202-024-00289-1).

Results

1. Consider adding visual elements, such as graphs or tables, to make the results more accessible and engaging.

Discussion

1. Expand the comparison of findings with international studies to highlight distinctive patterns observed in the Fujian population.

2. Elaborate on discrepancies between your results and previous studies, discussing potential reasons for these differences.

3. Provide practical implications of the findings, particularly focusing on policy recommendations and targeted interventions.

4. Explore the interaction between risk factors and protective factors influencing sleep quality in greater detail.

Limitations

1. Discuss the limitations of self-reported data more thoroughly, particularly with regard to response bias in sensitive topics like smoking.

2. Address the role of cultural differences in interpreting electronic product addiction and sleep quality.

Language and Formatting

1. Review the manuscript for grammatical errors and improve sentence structure, especially where sentences are overly lengthy or unclear.

2. Ensure consistent use of abbreviations, such as “PSQI,” throughout the text.

3. Standardize the formatting of tables and figures to enhance readability and professionalism.

Ethics and Data Transparency

1. Reiterate how ethical issues, such as anonymity and the potential biases associated with snowball sampling, were addressed.

2. Provide additional information about data availability and any plans for public access to the dataset.

Conclusions and Recommendations

1. Avoid making overly broad recommendations. Instead, suggest actionable interventions based on the findings, such as promoting reduced screen time before bed or implementing smoking cessation programs.

Reviewer #2: This manuscript is well-written and reflects a high standard of academic rigor. However, to further enhance the depth of the discussion section, the authors may consider incorporating insights from the following article: DOI: 10.1002/hsr2.70018. If deemed relevant, referencing this work could provide additional context or comparative perspectives, thereby enriching the overall quality of the manuscript.

6. PLOS authors have the option to publish the peer review history of their article (what does this mean? ). If published, this will include your full peer review and any attached files.

**Do you want your identity to be public for this peer review?** For information about this choice, including consent withdrawal, please see our Privacy Policy .

Reviewer #1: No

Reviewer #2: No

---

## [Author Response · Author response to Decision Letter 0]

24 Jan 2025

Dear Reviewers and Academic Editor Ahmad Neyazi:

Thank you for your letter and for the comments concerning our manuscript entitled "Sleep quality of college students in Fujian and its influencing factors: a cross-sectional study" (PONE-D-24-42285). Those comments are all valuable and very helpful for revising and improving our paper, as well as the important guiding significance to our research. We have studied the comments carefully and have made corrections which we hope meet with approval. Revised portions are marked in color on the paper. The main corrections in the paper and the responds to the comments are as following:

Responds to the Editor:

1. Please ensure that your manuscript meets PLOS ONE's style requirements, including those for file naming. The PLOS ONE style templates can be found at https://journals.plos.org/plosone/s/ file?id=wjVg/PLOSOne_formatting_sample_main_body.pdf and https://journals.plos.org/plosone /s/file?id=ba62/PLOSOne_formatting_sample_title_authors_affiliations.pdf

Response: We have revised the manuscript according to the PLOS ONE style templates and ensured that the file name is in compliance with the requirements.

The American Journal Experts (AJE) (https://www.aje.com/) is one such service that has extensive experience helping authors meet PLOS guidelines and can provide language editing, translation, manuscript formatting, and figure formatting to ensure your manuscript meets our· submission guidelines. Please note that having the manuscript copyedited by AJE or any other editing services does not guarantee selection for peer review or acceptance for publication.

Response: We have renamed and uploaded the files as requested.

3.Thank you for stating the following financial disclosure:  [This study was granted by the Natural Science Foundation of Quanzhou (Grant No.2021N124S PI: Fuzhi Liu)].  Please state what role the funders took in the study.  If the funders had no role, please state: "The funders had no role in study design, data collection and analysis, decision to publish, or preparation of the manuscript." If this statement is not correct you must amend it as needed. Please include this amended Role of Funder statement in your cover letter; we will change the online submission form on your behalf.

Response: We have made a new statement to the funders. (Please see page 14, line 328)

4. We note that your Data Availability Statement is currently as follows: [All relevant data are within the manuscript and its Supporting Information files.] Please confirm at this time whether or not your submission contains all raw data required to replicate the results of your study. Authors must share the “minimal data set” for their submission. PLOS defines the minimal data set to consist of the data required to replicate all study findings reported in the article, as well as related metadata and methods (https://journals.plos.org/plosone/s/data-availability#loc-minimal-data-set-definition).

Response: We have made new statements about data availability. (Please see page 15, line 344)

Responds to the reviewer’s comments:

Reviewer #1:

Introduction

1. Broaden the discussion on the importance of sleep quality by incorporating a global perspective and referencing recent studies.

Response: According to your revision suggestions, the revision has been made in the revised manuscript. (Please see page 2, line 63)

2. Strengthen the rationale for focusing on Fujian Province, particularly the relevance of electronic product addiction in this region.

Response: We have made changes in the revised version. (Please see page 3, line 76)

3. Include the following studies on adolescent sleep quality to enrich the introduction:

DOI: 10.1016/j.sleepe.2024.100102

DOI: 10.1016/j.sleepe.2024.100075

Response: We have added references in the revised manuscript. (Please see page 2, line 64)

Methodology

4. Provide a more detailed justification for using snowball sampling, considering its limitations in terms of representativeness and potential biases.

Response: Snowball sampling is a non-probability sampling technique that primarily involves the identification of additional sample participants from an initial cohort. This approach is generally cost-effective and practical. However, when the sample is restricted to individuals with similar characteristics, it may result in significant issues related to representativeness. Furthermore, the method's dependence on interpersonal networks can introduce certain biases and limitations in specific contexts. The survey in question was conducted in December 2022, during a period in China characterized by stringent COVID-19 prevention and control measures, which resulted in low levels of personnel mobility and rendered offline surveys impractical. Consequently, this study employed an online survey methodology, focusing on particular student demographics to expand the pool of research subjects.

5. Explain why the Mobile Phone Addiction Index (MPAI) was selected and how it is particularly suited for this study population.

Response: The Mobile Phone Addiction Index (MPAI) was created by Professor Leung Liang from the University of Hong Kong, drawing upon the diagnostic criteria for addiction outlined in the American Psychiatric Association's Diagnostic and Statistical Manual of Mental Disorders (4th Edition). This instrument comprises 17 items and is primarily utilized for the assessment of mobile phone addiction in adolescents and college students. The MPAI exhibits strong reliability and validity and has been extensively employed within the Chinese demographic. Furthermore, its effectiveness has been validated in international populations.

6. Include examples of psychometric validation of the Pittsburgh Sleep Quality Index (PSQI) from other countries (e.g., DOI: 10.1007/s44202-024-00289-1).

Response: We have added examples of psychometric validation of the Pittsburgh Sleep Quality Index (PSQI) from other countries in the article.

Results

7. Consider adding visual elements, such as graphs or tables, to make the results more accessible and engaging.

Response: The form has been restructured to enhance readability, while additional details can be found in the Supplementary Information. The data collected from this survey is diverse, making it challenging to represent visually.

Discussion

8. Expand the comparison of findings with international studies to highlight distinctive patterns observed in the Fujian population.

Response: As shown in the revised manuscript, we have now revised the Discussion section substantially and included comparisons with relevant international studies.(Please see page 10, line 205)

9. Elaborate on discrepancies between your results and previous studies, discussing potential reasons for these differences.

Response: We have already discussed the differences between the results of this study and previous studies in the revised manuscript.(Please see page 11, line 215)

10. Provide practical implications of the findings, particularly focusing on policy recommendations and targeted interventions.

Response: We have made reasonable modifications to the recommendations of this study in the revised version.(Please see page 13, line 294)

11. Explore the interaction between risk factors and protective factors influencing sleep quality in greater detail.

Response: We have already made a comprehensive discussion of the protective and risk factors for sleep quality in the revised draft.(Please see page 11, line 238)

Limitations

12. Discuss the limitations of self-reported data more thoroughly, particularly with regard to response bias in sensitive topics like smoking.

Response: We agree with the reviewer that self-reported data may be subject to recall bias. We now have acknowledged this weakness in the limitation section in the revised manuscript. (Please see page 13, line 285)

13. Address the role of cultural differences in interpreting electronic product addiction and sleep quality.

Response: This research primarily investigated the determinants of sleep quality, with a focus on cultural differences and electronic addiction. Nevertheless, the interplay between cultural differences and electronic addiction was not examined, and therefore, it is not addressed in the limitations section.

Language and Formatting

14. Review the manuscript for grammatical errors and improve sentence structure, especially where sentences are overly lengthy or unclear.

Response: We conducted a thorough review of the grammatical inaccuracies present in the manuscript and enhanced the sentence structure, particularly in instances where the sentences were excessively lengthy or ambiguous.

15. Ensure consistent use of abbreviations, such as “PSQI,” throughout the text.

Response: We thoroughly checked the abbreviations of the manuscript to ensure their consistency.

16. Standardize the formatting of tables and figures to enhance readability and professionalism.

Response: The statistical tables throughout the manuscript have been standardized to enhance both readability and professionalism.

Ethics and Data Transparency

17. Reiterate how ethical issues, such as anonymity and the potential biases associated with snowball sampling, were addressed.

Response: We have modified it in the ethical considerations section according to your revision opinion. (Please see page 4, line 126)

18. Provide additional information about data availability and any plans for public access to the dataset.

Response: We provide information about data availability and access to raw data at the end of this article. (Please see page 14, line 338)

Conclusions and Recommendations

19. Avoid making overly broad recommendations. Instead, suggest actionable interventions based on the findings, such as promoting reduced screen time before bed or implementing smoking cessation programs.

Response: We have revised the effective and feasible recommendations in the article. (Please see page 13, line 294)

Special thanks to you for your good comments.

Reviewer #2: This manuscript is well-written and reflects a high standard of academic rigor. However, to further enhance the depth of the discussion section, the authors may consider incorporating insights from the following article: DOI: 10.1002/hsr2.70018. If deemed relevant, referencing this work could provide additional context or comparative perspectives, thereby enriching the overall quality of the manuscript.

Response: On your advice, we have adopted the views of this article to enrich the quality of our manuscript. (Please see page 12, line 276)

Special thanks to you for your good comments.

We tried our best to improve the manuscript and made some changes. These changes will not influence the content and framework of the paper. Moreover, here we did not list the changes but marked them in red in the revised paper.

We appreciate your warm work earnestly and hope that the correction will be approved.

Once again, thank you very much for your comments and suggestions.

---

## [Decision Letter · Decision Letter 1]

31 Jan 2025

Sleep quality of college students in Fujian and its influencing factors: a cross-sectional study

PONE-D-24-42285R1

Dear Dr. Liu,

We’re pleased to inform you that your manuscript has been judged scientifically suitable for publication and will be formally accepted for publication once it meets all outstanding technical requirements.

Kind regards,

Ahmad Neyazi

Academic Editor

PLOS ONE

Additional Editor Comments (optional):

Thank you for revising the manuscript. I am pleased to accept the current version based on the reviewers' comments.

Reviewers' comments:

Reviewer's Responses to Questions

**Comments to the Author**

1. If the authors have adequately addressed your comments raised in a previous round of review and you feel that this manuscript is now acceptable for publication, you may indicate that here to bypass the “Comments to the Author” section, enter your conflict of interest statement in the “Confidential to Editor” section, and submit your "Accept" recommendation.

Reviewer #1: All comments have been addressed

Reviewer #2: All comments have been addressed

2. Is the manuscript technically sound, and do the data support the conclusions?

Reviewer #1: Yes

Reviewer #2: Yes

3. Has the statistical analysis been performed appropriately and rigorously? 

Reviewer #1: Yes

Reviewer #2: Yes

4. Have the authors made all data underlying the findings in their manuscript fully available?

Reviewer #1: Yes

Reviewer #2: Yes

5. Is the manuscript presented in an intelligible fashion and written in standard English?

Reviewer #1: Yes

Reviewer #2: Yes

6. Review Comments to the Author

Reviewer #1: Thank you for addressing all the comments. I think the current version of the manuscript is acceptable to be published in PLOS One.

Reviewer #2: I extend my appreciation to the authors for their commendable work. The revised manuscript meets the standards for publication.

7. PLOS authors have the option to publish the peer review history of their article (what does this mean? ). If published, this will include your full peer review and any attached files.

**Do you want your identity to be public for this peer review?** For information about this choice, including consent withdrawal, please see our Privacy Policy .

Reviewer #1: **Yes: ** Abdul Qadim Mohammadi

Reviewer #2: **Yes: ** Nosaibah Razaqi

---

## [Editor Report · Acceptance letter]

PONE-D-24-42285R1

PLOS ONE

Dear Dr. Liu,

I'm pleased to inform you that your manuscript has been deemed suitable for publication in PLOS ONE. Congratulations! Your manuscript is now being handed over to our production team.

Kind regards,

on behalf of

Dr. Ahmad Neyazi

Academic Editor

PLOS ONE